# A hepatocyte-specific transcriptional program driven by Rela and Stat3 exacerbates experimental colitis in mice by modulating bile synthesis

Jyotsna[1], Binayak Sarkar[1], Mohit Yadav[1], Alvina Deka[2], Manasvini Markandey[3], Priyadarshini Sanyal[4], Perumal Nagarajan[1], Nilesh Gaikward[5], Vineet Ahuja[3], Debasisa Mohanty[1], Soumen Basak[2], Rajesh S Gokhale[1,6]*

[1]Immunometabolism Laboratory, National Institute of Immunology, New Delhi, India; [2]System Immunology Laboratory, National Institute of Immunology, New Delhi, India; [3]Department of GastroEnterology, All India Institute of Medical Sciences, New Delhi, India; [4]Center for Cellular and Molecular Biology, Hyderabad, India; [5]Gaikwad Steroidomics Lab LLC, Davis, United States; [6]Department of Biology, Indian Institute of Science Education and Research, Pashan, India

*For correspondence:
rsg@nii.ac.in

**Abstract** Hepatic factors secreted by the liver promote homeostasis and are pivotal for maintaining the liver-gut axis. Bile acid metabolism is one such example wherein, bile acid synthesis occurs in the liver and its biotransformation happens in the intestine. Dysfunctional interactions between the liver and the intestine stimulate varied pathological outcomes through its bidirectional portal communication. Indeed, aberrant bile acid metabolism has been reported in inflammatory bowel disease (IBD). However, the molecular mechanisms underlying these crosstalks that perpetuate intestinal permeability and inflammation remain obscure. Here, we identify a novel hepatic gene program regulated by Rela and Stat3 that accentuates the inflammation in an acute experimental colitis model. Hepatocyte-specific ablation of Rela and Stat3 reduces the levels of primary bile acids in both the liver and the gut and shows a restricted colitogenic phenotype. On supplementation of chenodeoxycholic acid (CDCA), knock-out mice exhibit enhanced colitis-induced alterations. This study provides persuasive evidence for the development of multi-organ strategies for treating IBD and identifies a hepatocyte-specific Rela-Stat3 network as a promising therapeutic target.

## eLife assessment

The current version of the study presents **important** findings on how the RelA/Stat3-dependent gene program in the liver influences intestinal homeostasis. The evidence supporting the conclusions is **solid**, with new data added compared to an earlier version of the study. The work will be of interest to scientists in gastrointestinal research fields.

## Introduction

Under physiological conditions, the liver is continuously exposed to gut-derived antigens, which are either derived from the food we consume or are the product of microbial metabolism (*Tripathi et al., 2018*). The continuous interaction of extraneous antigens with the liver tissue makes it a tolerogenic organ, thereby justifying its unique anatomical location (*Horst et al., 2016*). However, during

intestinal inflammation or dysbiosis, as seen in conditions like IBD, there is an unregulated exchange of molecules across the gut vascular barrier, this potentially rewires the local as well as hepatic immunological and metabolic milieu (*Kobayashi et al., 2022*; *Bailey et al., 2023*). Over the last decade, several studies have focussed on analyzing the mechanistic aspects of gut-liver crosstalk, which intend to develop successful multi-organ therapies (*Tilg et al., 2022*; *Kessoku et al., 2021*).

Clinically, IBD overlaps with hepatobiliary conditions, it has been reported that 30% of IBD cases have abnormal liver function tests and around 5% of them even develop chronic hepatobiliary diseases (*Cappello et al., 2014*; *Gaspar et al., 2021*). During colitis, the first responders to extraneous agents incoming from the leaky gut are the Kupffer cells (*Yamada et al., 2003*). These cells have been shown to switch from a pro to an anti-inflammatory state in response to signals received from the colitogenic gut (*Taniki et al., 2018*). Besides kuffer cells, the parenchymal cells, hepatocytes which occupy ~80% of the liver, also respond to these varied danger signals. Hepatocytes have a robust secretary machinery and are known to secrete a variety of factors like the complement proteins, clotting factors, hepatokines, bile acids, etc. which regulate local as well as distant organ functions (*Kwon et al., 2021*; *Zhu et al., 2021*). Among the secreted factors, albumin is the most abundantly produced serum protein, which is implicated in the maintenance of redox balance and is known to attenuate DSS-induced colitis (*Yang et al., 2021b*). Highlighting the importance of liver secretome, a recent study suggests that FNDC4 has the potential to reduce colonic inflammation by acting on the colonic macrophages (*Bosma et al., 2016*). Similarly, bile acids are also known to play a pivotal role in regulating mucosal immune responses. Primary bile acids produced by the liver are often related to an enhanced proinflammatory state and its turnover to secondary bile acids is considered to be a critical step in the maintenance of homeostasis (*Sun et al., 2021*). The primary bile acids have been reported to accumulate in the inflamed colon, suggesting some intriguing crosstalk between the gut and liver (*Zhou et al., 2014*). However, it is obscure, how the liver perceives signals due to the disease-mediated impaired gut barrier to rewire secretory machinery and tackle the enhanced endotoxins influx.

Through this study, we propose Rela and Stat3 as key responders of inflammatory signaling in the liver tissue in response to intestinal aberrations. We further define a colitis resistance model based on the liver-specific knockout animals and propose a Rela/Stat3-CYP enzyme-mediated elevation of primary bile acid leading to immune-mediated damage to the gut. Briefly, this study establishes the functional significance of hepatic Rela and Stat3 in intestinal inflammation and emphasizes the therapeutic importance of targeting multiorgan crosstalk in inflammatory diseases.

## Results
### Engagement of hepatic Rela and/or Stat3 pathways in murine colitis model

Several studies have shown that the gut-liver bidirectional communication is critical in both the establishment and progression of IBD. Towards identifying liver pathways that affect intestinal impairment during IBD, we firstl examined whether 2% dextran sodium sulfate (DSS)-induced acute experimental colitis results in changes in liver pathophysiology. Analysis of biochemical parameters on day 6 of DSS treatment showed no significant alterations in the levels of alanine aminotransferase (ALT), aspartate aminotransferase (AST), gamma-glutamyl transferase (GGT), and bilirubin (*Figure 1—figure supplement 1a*). Furthermore, histological studies of these treated liver tissues from C57BL/6 mice showed neither morphological differences nor any hepatocellular fibrosis (*Figure 1—figure supplement 1b*). Thus, in general, the liver functionalities are maintained and no major damage in the liver tissue occurs in the colitogenic mice model. We then performed global transcriptomic studies of the liver tissue from day 6 colitogenic mice. Unsupervised clustering of the transcriptome data by principal component analysis (PCA) segregated the treated and control samples on PC1 with a variance of 53% (*Figure 1—figure supplement 1c*). Immunological and metabolic pathways appeared to be enriched in the list of regulated pathways. LPS-mediated signaling in the liver was identified as among the key differentially expressed systems (*Figure 1A*). It is proposed that microbial components, including LPS from the gut, can reach the liver through portal blood, activating the hepatic immune response during colitis-mediated barrier impairment. Interestingly, Rela and Stat3 are two important transcription factors activated by the microbial LPS and these two pathways frequently converge to elicit protective

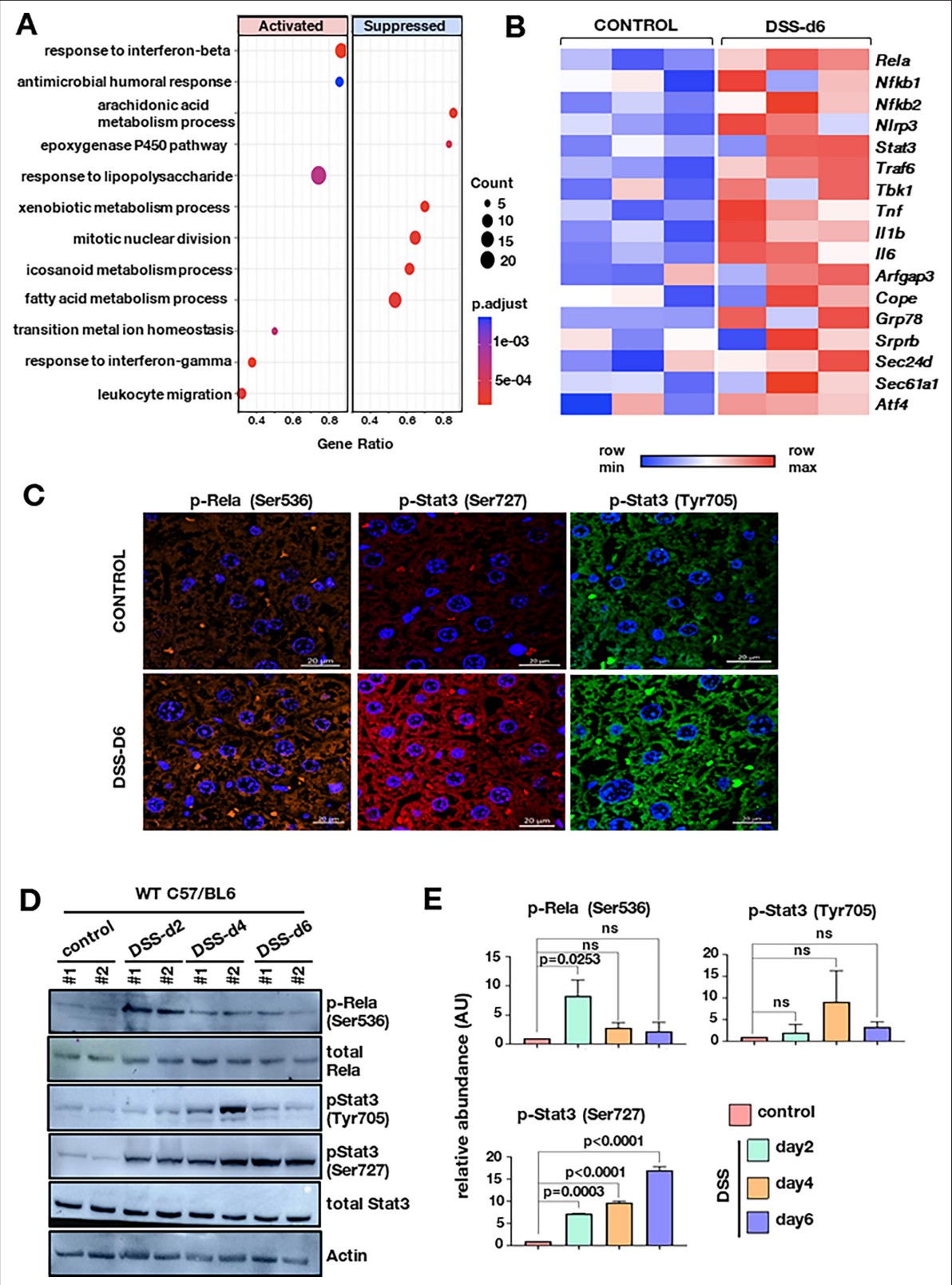

**Figure 1.** Initiation of colitis in mice leads to hepatic Rela and Stat3 activation. (**A**) GO pathway enrichment analysis was done for DEGs with adjusted p-value <0.05 on day 6 post DSS treatment. Bubble plot depicts the enrichment of pathways on day 6 for different genotypes, where the coordinate on the x-axis represents gene ratio, size of bubble represents the gene count and color represents the p-value. (**B**) Heatmap represents the normalized transcript count of the Rela and Stat3 pathway obtained from the RNA-seq experiment of three biological replicates. Scale is normalized across each

*Figure 1 continued on next page*

*Figure 1 continued*

row and color from blue to red represents minimum and the maximum values, respectively. (**C**) Representative confocal microscopy images show Rela and Stat3 activation in untreated and 6 days DSS-treated liver tissue of C57BL/6 mice. Images were taken at 40 X. Scale is 20 µm. (**D**) Western blot revealing the abundance of total Rela and total Stat3, and their phosphorylated functionally active forms, in the liver extracts prepared from wild-type C57BL/6 mice either left untreated or subjected to dextran sodium sulfate (DSS) treatment for 2, 4, and 6 days, respectively. (**E**) The signal intensity of bands corresponding to the indicated phosphorylated proteins was quantified from western blots, normalized against beta-actin, and presented as a bar plot. The data represent the means of three biological replicates ± SEM.

The online version of this article includes the following source data and figure supplement(s) for figure 1:

**Source data 1.** Table containing all the significantly regulated GO terms of which few are plotted in *Figure 1A*.

**Source data 2.** Data used for generating graph in *Figure 1E*.

**Source data 3.** Labeled and unedited blots shown in *Figure 1D*.

**Source data 4.** Zipped folder of the original blots shown in *Figure 1D*.

**Figure supplement 1.** Biochemical, histological and molecular characterization of mice liver upon colitis induction.

**Figure supplement 1—source data 1.** Data used for generating graph in *Figure 1—figure supplement 1a*.

**Figure supplement 1—source data 2.** Labeled and unedited blots shown in *Figure 1—figure supplement 1e*.

**Figure supplement 1—source data 3.** Zipped folder of the original blots in *Figure 1—figure supplement 1e*.

responses (*Balic et al., 2020*; *Sakai et al., 2017*; *Zhou et al., 2016*; *Ahyi et al., 2013*). Further analysis of the transcript abundance of Rela, Stat3, and other downstream genes reflected higher transcript abundance in the DSS-treated group as compared to the control (*Figure 1B*). Functional activation of Rela and Stat3 pathways requires phosphorylation of key residues, which we investigated in the liver sections of DSS-treated mice. Immunostaining studies showed an enhanced signal intensity of p-Rela (Ser536), p-Stat3 (Ser727), and p-Stat3 (Tyr705) in the treated liver sections, as compared to untreated samples (*Figure 1C*).

We then examined the time-dependent phosphorylation status of the two proteins of the liver post 0, 2, 4, and 6 days of DSS treatment. The RelA phosphorylation at Ser536 peaks at day 2 (about eightfold increase) following which a two to threefold increase was sustained till day 6 (*Figure 1D and E*). The Stat3 phosphorylation at Ser727 increases gradually from day 2 (eightfold) to day 6 (17-fold). Tyrosine phosphorylation, on the other hand, is transient and can be detected at day 4 (ninefold) post-treatment (*Figure 1D and E*).

Antibiotic-induced gut microbiome depletion has been frequently used to study gut microbiome roles in pathological conditions. Altered microbiota is also likely to modulate endotoxin influx. We, therefore, examined hepatic Rela/Stat3 activation after subjecting mice to oral administration of a broad-spectrum antibiotic cocktail (Amplicin, Neomycin, Metronidazole and Vancomycin) (*Hernández-Chirlaque et al., 2016*). Substantial decrease in bacterial load after 4 weeks was confirmed microbiologically. We induced colitis in mice with 2% DSS treatment along with antibiotic administration (*Figure 1—figure supplement 1d*). Previous data indicates that pStat3 (Ser727) is the most prominent marker that gets highly activated 6 days post DSS-treatment. Time course activation studies had indicated pStat3 (Ser727) as the most prominent marker to be activated 6 days post DSS-treatment; we examined the levels through western blotting. A marked reduction in pStat3 (Ser727) levels post-antibiotic treatment in the colitogenic wild-type mice (*Figure 1—figure supplement 1e*) could be noted, suggesting that optimal endotoxin flux may be a major driver of hepatic Rela/Stat3 signaling. Although it is important to note that antibiotic-induced microbiome depletion alters metabolic homeostasis by affecting gut signaling and colonic metabolism, in turn altering hepatic physiology (*Zarrinpar et al., 2018*).

## Hepatocyte-specific functions of Rela and Stat3 exacerbate experimental colitis

In order to discern the importance of transcriptional networks regulated by Rela and Stat3 we utilized a hepatocyte-specific knockout model. The Cre recombinase under albumin promoter drives the deletion of Rela and/or Stat3 in a tissue-specific manner (*Figure 2—figure supplement 1a and b*). To address the question, if ablation of Rela and Stat3 functioning in hepatocytes could modulate colonic inflammation, we subjected *Rela*Δhep, *Stat3*Δhep, *Rela*Δhep*Stat3*Δhep, and wild-type littermates

mice to acute DSS treatment. Previous studies suggest development of colitis in acute models is accompanied by shortening of the colon length, diarrhea, and rectal bleeding which is measured as the disease activity index (DAI) (*Chassaing et al., 2014*). Time course measurement of DAI in the wild-type showed an expected increase from day 3 (*Figure 2A*). While the DAI in Δhep mice was parallel to that observed in wild-type mice, *Stat3*Δhep mice displayed a subtle decrease in the DAI score, particularly on day 5. Surprisingly, *Rela*Δhep*Stat3*Δhep mice were almost resistant to induction of colitis, and only a minor increase in the DAI could be seen on day 6 post-onset of DSS treatment (*Figure 2A*). Concurrently, DSS treatment for 6 days showed substantial shortening of the colon length in wild-type, *Rela*Δhep, and *Stat3*Δhep mice, but not in *Rela*Δhep*Stat3*Δhep mice (*Figure 2B* and *Figure 2—figure supplement 1c*). Furthermore, we estimated the intestinal barrier permeability by measuring serum concentrations of fluorescein isothiocyanate (FITC)-dextran which was gavaged orally to these mice. A 12-fold increase in FITC-dextran in the serum of the wild-type animals was seen, while the *Rela*Δhep*Stat3*Δhep mice showed a substantially lower, fivefold increase (*Figure 2D*).

We also examined *Rela*Δhep*Stat3*Δhep mice for the epithelial architecture and mucin expression in the colon by following histological features and expression of colitogenic markers. DSS treatment led to extensive disruption of the intestinal crypts and depletion of the mucin layer in wild-type mice, accompanied by submucosal leukocyte infiltration. In contrast, *Rela*Δhep*Stat3*Δhep mice showed substantially attenuated epithelial changes along with minimal erosion of mucin layers (*Figure 2C* and *Figure 2—figure supplement 1*). RT-qPCR analysis revealed that DSS treatment of wild-type mice triggered a two-and-a-half-fold and twofold reduction in the colonic abundance of enterocyte markers, tight junction 1 (*Tjp1*), and occludin (*Ocln*), respectively (*Figure 2E*). Similarly, we noticed a fourfold and twofold decrease in the colonic abundance of goblet cell markers, mucins-2, and trefoil factor 3 (*Tff3*), respectively, in wild-type mice upon DSS treatment. Except for a moderate decrease in the *Tjp1* mRNA level, none of these mRNAs were downregulated upon DSS treatment in *Rela*Δhep*Stat3*Δhep mice. Taken together these studies reveal intriguing observations wherein the absence of Rela and Stat3 in hepatocytes protects mice from DSS-induced colitis.

## Induced hepatic expression of Rela and Stat3 stimulates primary bile acid synthesis pathway genes

To understand the role of the liver in imparting protective phenotype, exhibited by *Rela*Δhep*Stat3*Δhep strain, we performed an in-depth pathophysiological and molecular studies of the liver tissue of colitogenic knockout animals. Towards this, we evaluated the serum-based liver damage markers and histological features of the *Rela*Δhep*Stat3*Δhep mice. On day 4 and day 6 post DSS-treatment, ALT, AST, GGT, and bilirubin levels were within the physiological range, indicating no apparent damage to the liver (*Figure 3—figure supplement 1a*). Similarly, the histological studies also suggested no morphological differences in the DSS-treated *Rela*Δhep*Stat3*Δhep mice (*Figure 3—figure supplement 1b*). In conclusion, the above data indicates that the liver functions are maintained in the colitogenic *Rela*Δhep*Stat3*Δhep mice, similar to that observed in the wild-type animals. Furthermore, to dissect the mechanism underlying the resistant phenotype displayed by hepatocyte-specific *Rela*Δhep*Stat3*Δhep mice, we performed global transcriptome studies from liver tissues of treated/untreated *Rela*Δhep*Stat3*Δhep strain and their wild-type littermates. Unsupervised clustering of transcriptomic data using the PCA tool segregates wild-type and *Rela*Δhep*Stat3*Δhep mice samples subjected to DSS treatment (*Figure 3A*). Notably, untreated mice from the wild-type and *Rela*Δhep*Stat3*Δhep cluster together in the PCA plot. Comparative pathway enrichment analysis using GO terms for differentially expressed genes in the liver upon DSS treatment between wild-type and *Rela*Δhep*Stat3*Δhep mice showed differences in the acute phase responses, bile acid metabolic processes, response to ER stress, and one-carbon metabolism (*Figure 3B*). BA dysmetabolism has been reported in IBD patients by several studies and is also recapitulated in the mice models (*Zhou and Hylemon, 2014*; *Bromke and Krzystek-Korpacka, 2021*). The consensus thesis is that the levels of secondary BAs are lower, and primary BAs are elevated because of impairment of microbiota-mediated deconjugation and transformation activities. Concordantly, our analysis from colonic biopsies of IBD subjects also showed elevated levels of primary BAs (*Figure 3C* and *Figure 3—source data 4*).

BAs are synthesized in hepatocytes through the classical and alternate pathways catalyzed by a set of P450 enzymes (*Figure 3D*). Cholic acid (CA) and chenodeoxycholic acid (CDCA) are primary bile acids and are conjugated to either glycine (predominantly in humans) or taurine (mainly in mice).

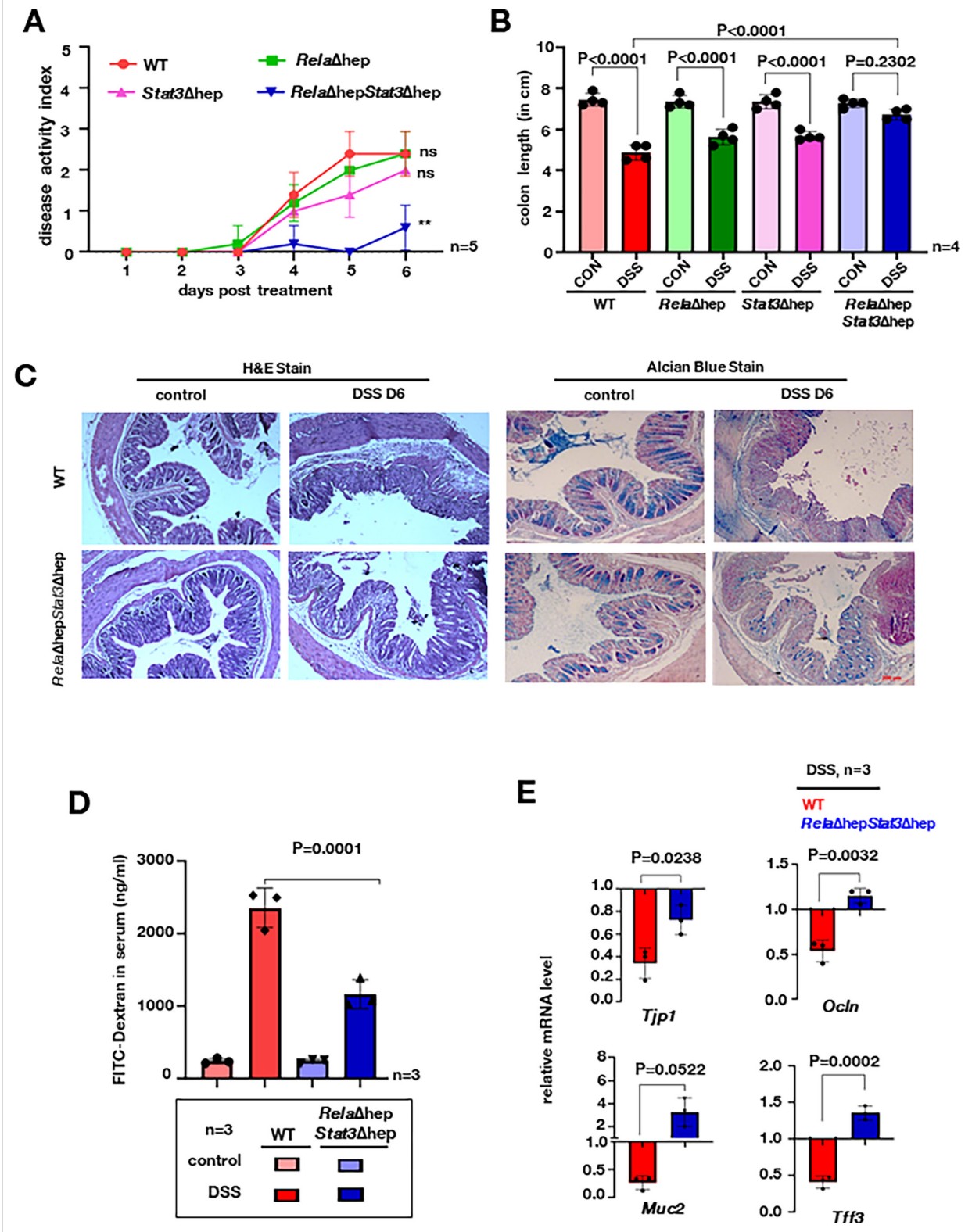

**Figure 2.** *Rela* and *Stat3* deficiency in hepatocytes ameliorates dextran sodium sulfate (DSS)-induced acute colitis in mice. (**A**) Line plot charting disease activity index of wild-type, *Rela*Δhep, *Stat3*Δhep, and *Rela*Δhep*Stat3*Δhep littermate mice subjected to treatment with 2% DSS for 6 days. (**B**) Bar plot depicting colon length measured on day 6 post-onset of DSS treatment of mice of the indicated genotypes. Untreated wild-type littermates of corresponding genotypes were used as controls. (**C**) Colon sections from untreated and DSS-treated mice of the indicated genotypes were examined for histological features involving hematoxylin and eosin (H&E) staining [left panel] and alcian blue staining [right panel]. Data were obtained in 10 X

*Figure 2 continued on next page*

*Figure 2 continued*

magnification and represented three experimental replicates; two fields per section and a total of three sections from each set were examined. (**D**) Bar plot quantifying gut permeability of untreated and DSS-treated wild-type and *Rela*Δhep*Stat3*Δhep mice. Briefly, the serum concentration of fluorescein isothiocyanate (FITC) was measured 6 hr after oral gavaging of the mice with FITC-dextran. (**E**) RT-qPCR reveals the relative abundance of the indicated mRNAs encoding broadly enterocyte-specific (above panel) or goblet cell-specific (below panel) markers in untreated or DSS-treated mice of the indicated genotypes.

The online version of this article includes the following source data and figure supplement(s) for figure 2:

**Source data 1.** Data used for generating graph in *Figure 2A*.

**Source data 2.** Data used for generating graph in *Figure 2B*.

**Source data 3.** Data used for generating graph in *Figure 2D*.

**Source data 4.** Data used for generating graph in *Figure 2E*.

**Figure supplement 1.** Methods for characterization of mice strain and analysis of colon pathology of colitogenic mice.

**Figure supplement 1—source data 1.** Labeled and unedited gel shown in *Figure 2—figure supplement 1a*.

**Figure supplement 1—source data 2.** Zipped folder of the original gel shown in *Figure 2—figure supplement 1a*.

**Figure supplement 1—source data 3.** Labeled and unedited blot shown in *Figure 2—figure supplement 1b*.

**Figure supplement 1—source data 4.** Zipped folder of the original blot shown in *Figure 2—figure supplement 1b*.

**Figure supplement 1—source data 5.** Data used for generating graph in *Figure 2—figure supplement 1d*.

Transcriptomics data strikingly showed downregulation of *Cyp7b1* in the *Rela*Δhep*Stat3*Δhep mice and the other biosynthesis enzymes also follow a similar trend (*Figure 3—figure supplement 1c–e*). Our RT-qPCR analyses substantiated that in comparison to DSS-treated wild-type mice, DSS-treated *Rela*Δhep*Stat3*Δhep mice expressed a reduced level of mRNAs encoding primary bile acid synthesis pathway enzymes *Cyp7a1*, *Cyp8b1*, *Cyp27a1,* and *Cyp7b1* in the liver (*Figure 3E*). Thus, our data propose a new hypothesis that hepatic Rela and Stat3 instruct a gene program in the liver of colitogenic mice that supports the expression of mRNAs encoding primary BA metabolism enzymes.

## Reducing the levels of primary bile acids dampened intestinal inflammation in colitogenic *Rela*Δhep*Stat3*Δhep mice

To investigate if the altered hepatic gene expression led to a change in the abundance of primary bile metabolites in *Rela*Δhep*Stat3*Δhep mice, we performed a targeted metabolomic study using LC-MS. Apart from measuring CA and CDCA, we also measured the level of CDCA-derived bile metabolites, namely ursodeoxycholic acid and α- and β-muricholic acid which are specifically produced in mice (*Honda et al., 2020*). Our analyses revealed the abundance of cholic acid in the liver, which was approximately sevenfold less in DSS-treated *Rela*Δhep*Stat3*Δhep mice compared to their wild-type counterparts (*Figure 4A*). Likewise, we captured a close to 10-fold decrease in the hepatic level of CDCA in DSS-treated knockout mice. A substantially reduced levels of ursodeoxycholic acid and α-muricholic acid. We further compared DSS-treated wild-type and *Rela*Δhep*Stat3*Δhep mice for the colonic abundance of these bile acids. Consistent with the levels observed in the liver, we found a significantly reduced accumulation of both CA and CDCA in the colon of DSS-treated *Rela*Δhep*Stat3*Δhep mice. The difference between wild-type and knockout mice was substantially more marked for CDCA (*Figure 4B*).

CDCA has been reported to engage the NLRP3 inflammasome, causing liver inflammation during cholestasis (*Gong et al., 2016*.) Furthermore, ex vivo experiments have shown that CDCA induces pro-inflammatory cytokine secretion by intestinal epithelial cells (*Horikawa et al., 2019*). Because, we noticed an altered abundance of these primary bile acids in *Rela*Δhep*Stat3*Δhep mice, we asked if DSS treatment dampened the pro-inflammatory response in these knockouts. Indeed, compared to DSS-treated wild-type mice, DSS-treated knockout mice presented with substantially reduced levels of mRNAs encoding key pro-inflammatory cytokines, namely Il1b, Tnfa, and Il6 (*Figure 4C*). Consistent with the less inflamed intestinal milieu, *Rela*Δhep*Stat3*Δhep mice also displayed a diminished frequency of inflammatory effector immune cells in the colitogenic gut (*Figure 4D* and *Figure 4—figure supplement 1a*). Confocal microscopy-based analyses of stained colon sections depicted a close to fivefold reduction in the recruitment of Ly6G+ cells, including neutrophils and monocytes, F4/80+ macrophages, and CD11c+ including macrophages and the dendritic cells in DSS-treated

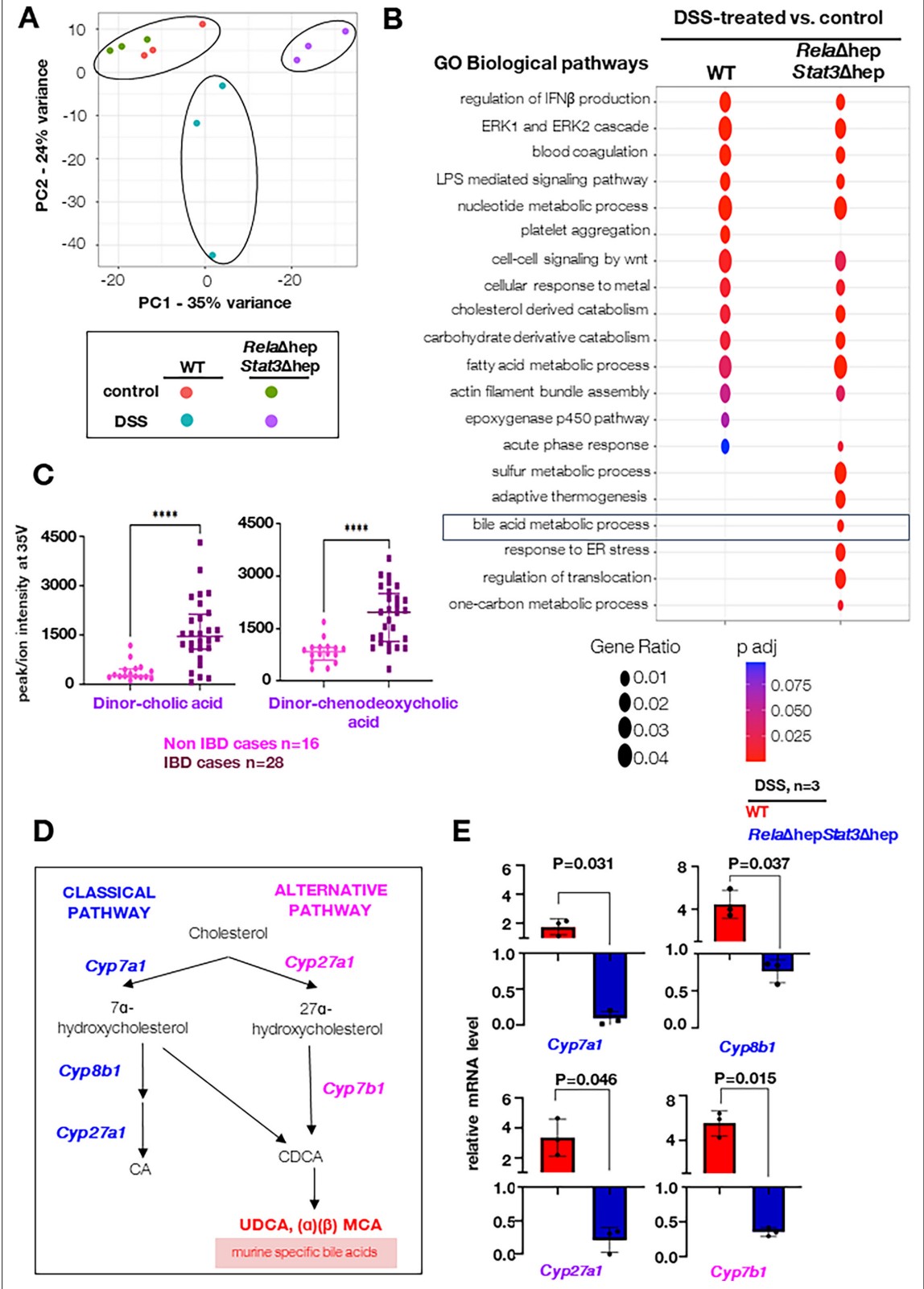

**Figure 3.** Charting hepatic gene expressions in colitogenic wild-type and *Rela*Δhep*Stat3*Δhep mice. (**A**) Principal component analysis (PCA) plot illustrating the hepatic transcriptome, identified through global RNA-seq analyses, of untreated or dextran sodium sulfate (DSS)-treated wild-type and *Rela*Δhep*Stat3*Δhep mice (n=3). DSS treatment was carried out for 6 days. (**B**) Bubble plot depicting the relative enrichment of GO biological terms for differentially expressed genes in wild-type or *Rela*Δhep*Stat3*Δhep mice. The gene ratio for a given term and the adjusted p-value associated with

*Figure 3 continued on next page*

*Figure 3 continued*

the enrichment score has been presented for the individual genetic backgrounds. (**C**) Dot plot of dinor-cholic acid and dinor-chenodeoxycholic acid as detected in an untargeted LC-MS-based quantification of bile acids in the mucosal biopsy samples from inflammatory bowel disease (IBD) and Non-IBD patients. (**D**) Schematic presentation of classical and alternate pathways of bile synthesis in mice liver tissue. CA, CDCA, MCA, and UDCA represent cholic, chenodeoxycholic, muricholic, and ursodeoxycholic acids, respectively. (**E**) RT-qPCR analyses comparing the hepatic abundance of indicated mRNAs encoding enzymes involved in bile metabolism in DSS-treated wild-type and *RelaΔhepStat3Δhep* mice (n=3). Fold change is relative to their corresponding wild-type littermates.

The online version of this article includes the following source data and figure supplement(s) for figure 3:

**Source data 1.** Table containing all the significantly regulated GO terms of which few are plotted in *Figure 3B*.

**Source data 2.** Data used for generating graph in *Figure 3C*.

**Source data 3.** Data used for generating graph in *Figure 3E*.

**Source data 4.** Table containing the demography of the control and UC patient for which the bile acids have been quantitated.

**Figure supplement 1.** Analysis of biochemical and molecular parameters of colitogenic wildtype and *RelaΔhepStat3Δhep* mice.

**Figure supplement 1—source data 1.** Data used for generating graph in *Figure 3—figure supplement 1a*.

knockout mice, compared to their wild-type counterparts (*Figure 4D* and *Figure 4—figure supplement 1a*). Taken together, we propose that a reduced accumulation of primary bile acids alleviates experimental colitis in *RelaΔhepStat3Δhep* mice.

## Supplementing chenodeoxycholic acid restores colitogenic sensitivity of *RelaΔhepStat3Δhep* mice

Recent studies have highlighted the role of CA in potentiating intestinal damage by impairing Lgr5[+] intestinal stem cells. However, the role of CDCA in regulating intestinal inflammation in colitogenic mice remains unclear. Therefore, we supplemented CDCA intraperitoneally daily for 6 days during DSS treatment in the wild-type or *RelaΔhepStat3Δhep* mice and scored the DAI (*Figure 5—figure supplement 1a*). The optimum concentration of CDCA was determined by performing a dose-response curve. Mere daily supplementation of CDCA at the concentration of 10 mg/kg of body weight, elevated the levels of CDCA in the colon (*Figure 5—figure supplement 1b*) without inducing any colitogenic phenotype in either wild-type or *RelaΔhepStat3Δhep* mice even after 6 days (*Figure 5—figure supplement 1c–e*). Moreover, the DAI of wild-type mice treated with both DSS and CDCA showed no change (*Figure 5A*). On the other hand, CDCA supplementation of the DSS-treated *RelaΔhepStat3Δhep* mice showed DAI equivalent to wild-type DSS-treated mice with concomitant supplementation of colon length shortening (*Figure 5A and B*). Histological analyses of DSS-treated colon sections further substantiated that CDCA supplementation together with DSS treatment was sufficient for imparting damage to the colonic crypts of the colitis-resistant *RelaΔhepStat3Δhep* mice (*Figure 5C*). Consistent with the observed intestinal pathologies, bile acid supplementation also triggered a drastic reduction in the colonic abundance of mRNAs encoding the junctional proteins *Tjp1* and *Ocln* as well as *Muc2* in DSS-treated *RelaΔhepStat3Δhep* mice (*Figure 5D*). Concurrently, bile acid-supplemented *RelaΔhepStat3Δhep* mice show prominently upregulated expression of genes encoding the pro-inflammatory cytokines *Il1b*, *Tnfa*, and *Il6* upon DSS treatment, compared to those treated with DSS alone (*Figure 5E*). Thus, we conclude that RelA and Stat3-driven accumulation of CDCA aggravates intestinal inflammation-induced damage during experimental colitis.

## Discussion

IBD is a heterogeneous group of chronic inflammatory intestinal disorders that are influenced by environmental cues, intestinal dysbiosis, and the local immune responses (*Ananthakrishnan et al., 2018*; *de Souza and Fiocchi, 2016*). Chronic intestinal inflammation remodels the permeability barrier resulting in leakage of microbial components, including LPS, through the portal circulation thereby engaging extraintestinal organs such as the liver. The conventional treatment regimen for IBD involves aminosalicylates, corticosteroids, and anti-TNF agents to counter bowel inflammation (*Gómez-Gómez et al., 2015*). Systemic effects are only addressed upon worsening of symptoms (*Veloso, 2011*). To improve the clinical management of IBD, there is a need to identify targets and mechanisms that can concurrently address other associated causes. In this study, we reveal the surprising role of hepatic

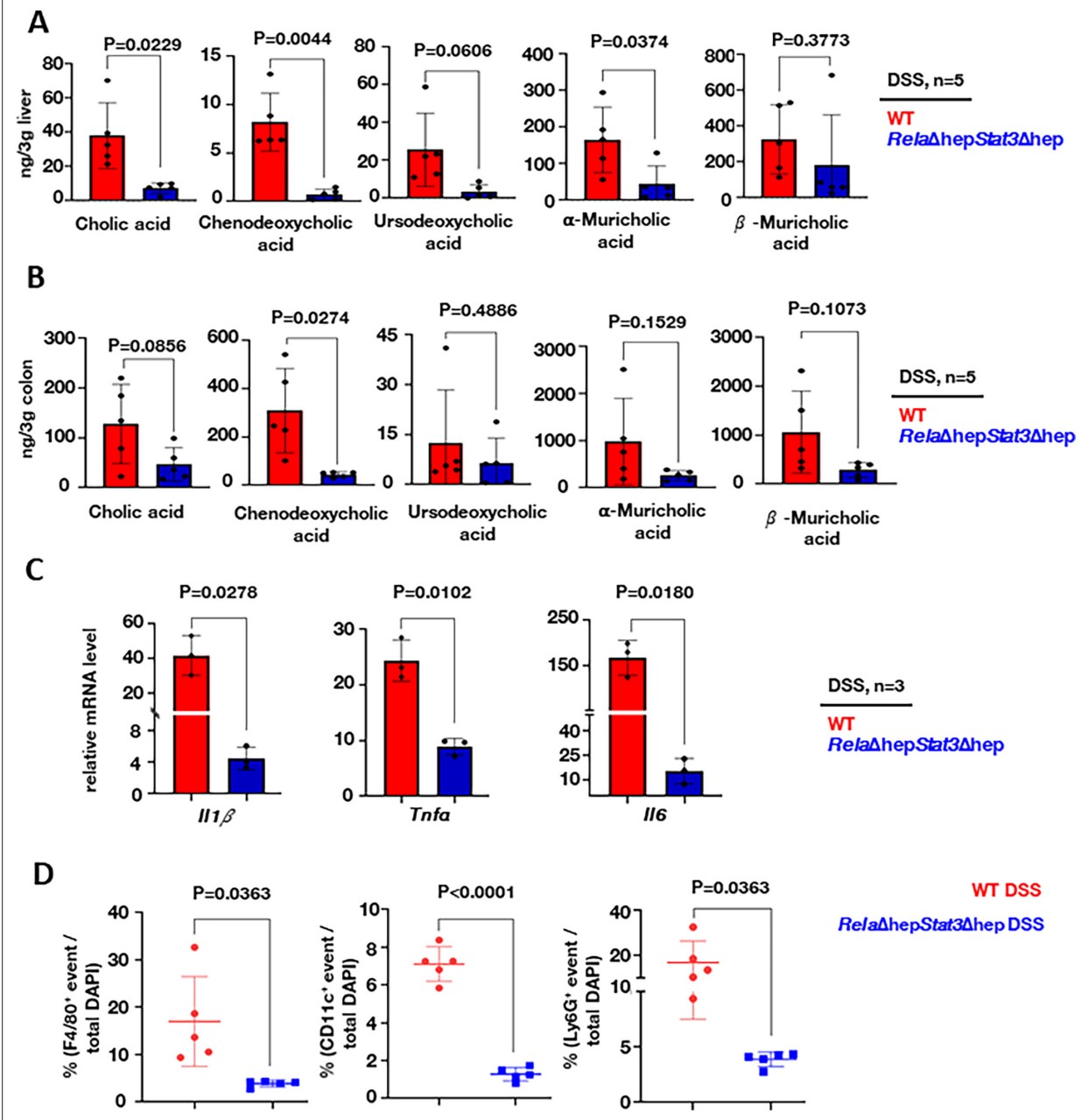

**Figure 4.** Altered accumulation of primary bile acids in *Rela*Δhep*Stat3*Δhep mice accompanies a less severe inflammatory signature in the colitogenic gut. Targeted LC-MS-based quantification of primary bile acid in the liver (**A**) and the colon (**B**) of dextran sodium sulfate (DSS)-treated wild-type and *Rela*Δhep*Stat3*Δhep mice (n=5). (**C**) RT-qPCR analyses comparing the colonic abundance of indicated mRNAs encoding pro-inflammatory cytokines (n=3) for DSS-treated wild-type and *Rela*Δhep*Stat3*Δhep mice. Fold change is relative to their corresponding wild-type littermates. (**D**) Dot-plot representing the frequency of F4/80+, CD11c+, and Ly6G+ cells among total DAPI-stained cells in the colon sections derived from DSS-treated wild-type and *Rela*Δhep*Stat3*Δhep mice.

The online version of this article includes the following source data and figure supplement(s) for figure 4:

**Source data 1.** Data used for generating graph in *Figure 4A*.

**Source data 2.** Data used for generating graph in *Figure 4B*.

**Source data 3.** Data used for generating graph in *Figure 4C*.

**Source data 4.** Data used for generating graph in *Figure 4D*.

**Figure supplement 1.** Microscopic analysis of immune cell infilteration in the colon of colitogenic mice.

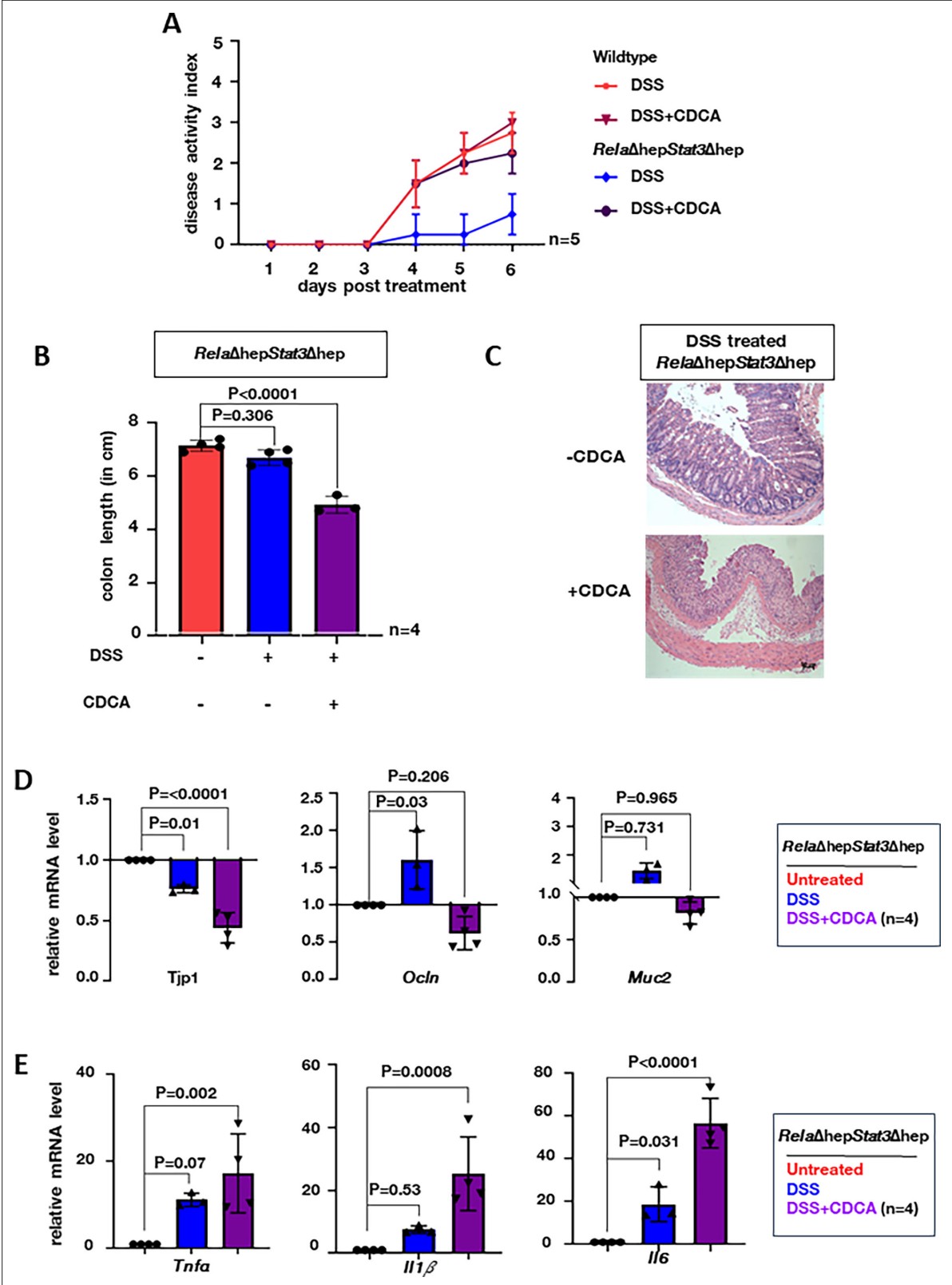

**Figure 5.** Supplementing chenodeoxycholic acid (CDCA) restores the colitogenic sensitivity in *RelaΔhepStat3Δhep* mice. (**A**) Line plot charting the disease activity in a time course of wild-type and *RelaΔhepStat3Δhep* mice subjected to dextran sodium sulfate (DSS) treatment while being supplemented with 10 mg/kg CDCA daily. Mice devoid of CDCA supplementation were treated with DMSO as controls. (**B**) Bar plot comparing the colon length of *RelaΔhepStat3Δhep*mice subjected to DSS treatment for 6 days in the absence or presence of CDCA supplementation. (**C**) Hematoxylin

*Figure 5 continued on next page*

*Figure 5 continued*

and eosin (H&E) stained colon sections from DSS-treated *Rela*Δhep*Stat3*Δhep mice with and without CDCA supplementation. Data were obtained in 10 X magnification, this is a representative of three experimental replicates and a total of four sections from each set were examined. RT-qPCR analyses comparing the colonic abundance of indicated mRNAs encoding (**D**) IEC-specific markers and (**E**) pro-inflammatory cytokines in mice subjected to DSS treatment for 6 days in the absence or presence of CDCA supplementation (n=4). Untreated *Rela*Δhep*Stat3*Δhep mice were used as controls.

The online version of this article includes the following source data and figure supplement(s) for figure 5:

**Source data 1.** Data used for generating graph in *Figure 4A*.

**Source data 2.** Data used for generating graph in *Figure 4B*.

**Source data 3.** Data used for generating graph in *Figure 4D*.

**Source data 4.** Data used for generating graph in *Figure 4E*.

**Figure supplement 1.** Supplementation of Chenodeoxycholic acid (CDCA) to the mice and its effect on gut.

**Figure supplement 1—source data 1.** Data used for generating graph in *Figure 5—figure supplement 1b*.

**Figure supplement 1—source data 2.** Data used for generating graph in *Figure 5—figure supplement 1d*.

Rela-Stat3 in regulating the biosynthesis of primary bile acids, whose activation augments intestinal inflammation.

Lipopolysaccharides (LPS), one of the key components of microbial products, is known to induce inflammasome assembly by NF-κB/Lipocalin2-dependent axis in macrophages of colitogenic mice (*Kim et al., 2022*). Similarly, LPS-mediated non-canonical Stat3 activation through TLR4 ligand reprograms metabolic and inflammatory pathways in macrophages (*Balic et al., 2020*). In the acute DSS-induced colitis model, the release of LPS and other endotoxins would first activate kupffer cells of the liver, which then progressively triggers a response in relatively quiescent hepatocytes (*Taniki et al., 2018*). We argued that Stat3-NF-κB signaling may be of specific significance since these pathways in hepatocytes are known to control the secretion of hepatic factors. We indeed observed activation of the canonical arm of the NF-κB signaling pathway with RelA phosphorylation at Ser536 residue following DSS treatment. Interestingly, non-canonical Stat3 activation was detected with Ser727 phosphorylation preceding the phosphorylation of canonical Tyr705 residue. To understand the significance of these signaling pathways of hepatocytes in the context of disease outcomes and its functional consequence in disease management, we subjected the single and double hepatocyte-specific knockout mice to DSS treatment. Surprisingly, the *Rela*Δhep*Stat3*Δhep mice had attenuated disease activity index as well as diminished inflammatory response in the gut. Previous studies wherein attenuation of the colitogenic phenotype has been reported are for the gut-resident cells (*Chawla et al., 2021*; *Shi et al., 2020*; *Weisser et al., 2011*; *Bessman et al., 2020*). This study is amongst the first such, where ablation of genes in a distal organ - the liver - ameliorates colitis.

We then investigated the mechanisms underlying signaling networks in the liver that promote the colitis phenotype. Previously, hepatic Rela and Stat3 have been reported to act cooperatively to render protection against pulmonary infection, by eliciting an acute phase response, which enhances the pulmonary immune response (*Quinton et al., 2012*; *Hilliard et al., 2015*). We observe similar hepatic functions of Rela and Stat3 that amplify the immune response during experimental colitis, aggravating intestinal pathologies. Our unbiased transcriptome analysis of the liver genes revealed several metabolic and immune pathways to be altered in colitogenic mice. Amongst these, the biosynthesis of primary BAs prompted our immediate attention. The primary bile acids are known to have inflammatory activity and the cardinal paradigm in IBD is the decreased levels of secondary bile acid (*Sinha et al., 2020*; *Yang et al., 2021a*). This has been attributed to the altered gut microflora that are known to convert primary BAs into secondary BAs. It is important to note that most of these studies are carried out with fecal samples and a recent study that profiled the human intestinal environment under physiological conditions concluded that stool-based measurements do not reflect the true composition of BAs along the intestinal tracts. Intestinal samples were dominated by primary BAs, whereas stool samples were dominated by secondary BAs (*Shalon et al., 2023*). We, therefore, measured the intestinal BAs in IBD subjects by using mucosal biopsies and observed elevated levels of primary BAs in the IBD subjects as compared to non-IBD subjects.

Analysis of colitogenic mice intestinal samples also revealed enhanced primary BAs using mass spectrometry-based measurements. We observe both CDCA and CA accumulate in the intestine and liver of colitogenic wild-type mice when compared to knockout littermates. This accumulation of

primary BA in both the intestine and liver is likely to be a consequence of increased biosynthesis in the liver and subsequent transport to the gut. Indeed, multiple primary bile synthesis pathway genes namely *Cyp7a1*, *Cyp27a1*, *Cyp7b1*, and *Cyp8b1* were substantially upregulated in the liver. Our observation is contradictory to the recently published work by *Gui et al., 2023* which suggests that during IBD there is a downregulation of the biosynthesis machinery leading to reduced bile levels. Currently, we assume that these differences may arise due to the dose of DSS, the time course of the experiment, and the mice strain adapted for different studies leading to the difference in the transcriptional and metabolic landscape during IBD (*Gui et al., 2023*; *Chen et al., 2022*; *Zhou et al., 2014*).

Conventionally the cyp450 enzymes involved in the bile acid biosynthesis pathway are known to be regulated by the FXR signaling cascade (*Eloranta and Kullak-Ublick, 2008*). Our data suggest that RelA and Stat3 transcriptionally control these pathways under stress conditions. These two transcription factors have been known to collaborate in a variety of physiological and disease settings. Further studies would elucidate a mechanistic understanding of how these hepatocyte-specific transcriptional regulatory circuitry drives hepatic bile synthesis during gastrointestinal abnormalities. Mutant mice showed refractory behavior towards colitogensis, where mere supplementation of CDCA resulted in exacerbated colitis phenotype in the gut. Previous studies of increased CA have been attributed to

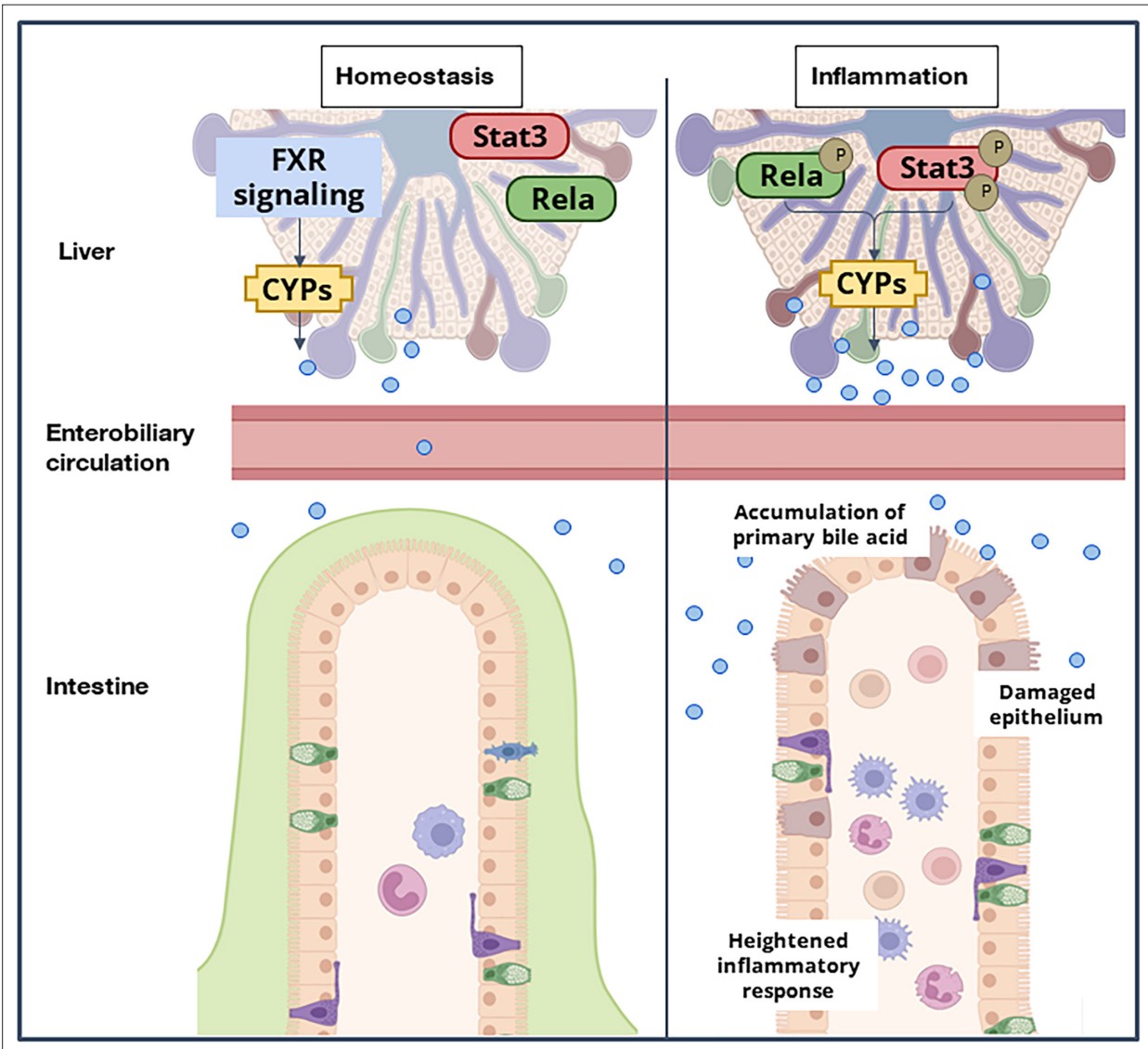

**Figure 6.** A model depicting the immuno-metabolic network linking the inflammation-induced hepatic signaling pathway to intestinal pathologies in mice.

colitis and CA was shown to limit the self-renewal capacity of intestinal stem cells leading to impaired intestinal restitution (*Chen et al., 2022*). We now propose that increased synthesis and secretion of the primary bile acids orchestrates intestinal inflammation. It is increasingly important to understand how bile acid signaling networks are affected in distinct organs where the bile acid composition differs, and how these networks impact intestinal diseases. Our studies identify a new, important Rela-Stat3 network system of hepatocytes that could enable the development of therapeutics that target BA imbalance by suppressing host-specific stress-induced pathways (*Figure 6*).

Until now, only immunosuppressive agents and immunomodulators have been conventionally considered as therapeutic measures to manage IBD. However, with increasing research on the role of hepatic bile acid metabolism during experimental colitis, its potential cannot be undermined in the clinical setting. The potential of bile acids as a therapeutic target has been harnessed in the past; bile acid sequestrants have been utilized as a treatment for hyperlipidemia (*Camilleri and Gores, 2015*). Remedies like fecal microbial transplantation, which serve to normalize the bile acid ratios in the gut, have emerged as potential therapeutics in the last decade for IBD (*Paramsothy et al., 2019*, *Sinha et al., 2020*). However, the potential of altering hepatic bile metabolism has remained unexplored for IBD, possibly due to a lack of mechanistic insight. Towards this, our work demonstrates the proinflammatory potential of CDCA during colitis following the activation of the Rela/Stat3 pathway. The suppression of Rela/Stat3-induced CDCA could provide beneficial effects in IBD patients while protecting the basal bile acid levels (through FXR signaling). Thus our studies identify a hepatocyte-specific Rela/Stat3 network as a potential therapeutic target for intestinal diseases. Another approach could be the use of bile acid sequestrants, which will temporarily decrease the levels of primary bile acids in the colon until the proinflammatory pathways are dampened as a combinatorial therapy alongside existing treatments.

## Key resources table

| Reagent type (species) or resource | Designation | Source or reference | Identifiers | Additional information |
|---|---|---|---|---|
| Strain, strain background (*Mus musculus*) | *Alb-Cre Rela^{-/-} Stat3^{-/-}* | A gift from Dr. Lee Quinton's lab at Boston University | | C57BL/6 |
| Strain, strain background (*Mus musculus*) | *Rela^{fl/fl} Stat3^{fl/fl}* | A gift from Dr. Lee Quinton's lab at Boston University | | C57BL/6 |
| Strain, strain background (*Mus musculus*) | *Alb-Cre Rela^{-/-}* | A gift from Dr. Lee Quinton's lab at Boston University | | C57BL/6 |
| Strain, strain background (*Mus musculus*) | *Rela^{fl/fl}* | A gift from Dr. Lee Quinton's lab at Boston University | | C57BL/6 |
| Strain, strain background (*Mus musculus*) | *Alb-Cre Stat3^{-/-}* | A gift from Dr. Lee Quinton's lab at Boston University | | C57BL/6 |
| Strain, strain background (*Mus musculus*) | *Stat3^{fl/fl}* | A gift from Dr. Lee Quinton's lab at Boston University | | C57BL/6 |
| Antibody | anti-mouse Ly-6G-PE (Rat monoclonal) | BD Biosciences | Cat# 551461, RRID: AB_394208 | IF (1:500) |
| Antibody | anti-mouse F4/80-FITC (Mouse Polyclonal) | BioLegend | Cat# 157309, RRID: AB_2876535 | IF (1:500) |
| Antibody | anti-mouse CD11c-PE (Hamster monoclonal) | BioLegend | Cat# 117308, RRID: AB_313777 | IF (1:500) |
| Antibody | anti-mouse CD4-PE (Rat monoclonal) | BioLegend | Cat# 100408, RRID: AB_312693 | IF (1:500) |

*Continued on next page*

*Continued*

| Reagent type (species) or resource | Designation | Source or reference | Identifiers | Additional information |
|---|---|---|---|---|
| Antibody | anti-mouse STAT3 (mouse monoclonal) | ThermoFisher Scientific | Cat# MA1-13042, RRID: AB_10985240 | IF (1:100) WB (1:1000) |
| Antibody | anti-mouse Rela (Rabbit polyclonal) | Santa Cruz Biotechnology | Cat# sc372, RRID: AB_632037 | IF (1:100) WB (1:1000) |
| Antibody | anti-mouse Phospho-Stat3 (Ser727) (Rabbit monoclonal) | Cell Signaling Technology | Cat# 34911, RRID: AB_2737598 | IF (1:100) WB (1:1000) |
| Antibody | anti-mouse Phospho-Stat3 (Ser727) (Rabbit monoclonal) | Cell Signaling Technology | Cat# 34911, RRID: AB_2737598 | IF (1:100) WB (1:1000) |
| Antibody | anti-mouse Phospho-Stat3 (Tyr705) (Rabbit monoclonal) | Cell Signaling Technology | Cat# 34911, RRID: AB_2737598 | IF (1:100)tjp WB (1:1000) |
| Antibody | anti-mouse Phospho-NF-κB p65 (Ser536) (Rabbit polyclonal) | Cell Signaling Technology | Cat# 3031, RRID: AB_330559 | WB (1:1000) |
| Antibody | anti-mouse GAPDH (Rabbit monoclonal) | Cell Signaling Technology | Cat# 2118, RRID: AB_561053 | WB (1:1000) |
| Antibody | anti-mouse β-Actin (Rabbit polyclonal) | Cell Signaling Technology | Cat# 4967, AB_330288 | WB (1:1000) |
| Antibody | anti-rabbit IgG (H+L) Secondary Antibody Alexa Fluor Plus 555 (Goat polyclonal) | Thermo Fisher Scientific | Cat# A32732, AB_2633281 | IF (1:2000) |
| Sequence-based reagent | *Tjp1_F* | This paper | PCR primer | GCTTTAGCGAACAGAAGGAGC |
| Sequence-based reagent | *Tjp1_R* | This paper | PCR primer | TTCATTTTTCCGAGACTTCACCA |
| Sequence-based reagent | *Ocln_F* | This paper | PCR primer | TGAAAGTCCACCTCCTTACAGA |
| Sequence-based reagent | *Ocln_R* | This paper | PCR primer | CCGGATAAAAAGAGTACGCTGG |
| Sequence-based reagent | *Muc2_F* | This paper | PCR primer | AGGGCTCGGAACTCCAGAAA |
| Sequence-based reagent | *Muc2_R* | This paper | PCR primer | CCAGGGAATCGGTAGACATCG |
| Sequence-based reagent | *Tff3_F* | This paper | PCR primer | TTGCTGGGTCCTCTGGGATAG |
| Sequence-based reagent | *Tff3_R* | This paper | PCR primer | TACACTGCTCCGATGTGACAG |
| Sequence-based reagent | *Il1b_F* | This paper | PCR primer | CATCCCATGAGTCACAGAGGATG |
| Sequence-based reagent | *Il1b_R* | This paper | PCR primer | ACCTTCCAGGATGAGGACATGAG |
| Sequence-based reagent | *Tnf_F* | This paper | PCR primer | CTGAACTTCGGGGTGATCGG |
| Sequence-based reagent | *Tnf_R* | This paper | PCR primer | GGCTTGTCACTCGAATTTTGAGA |
| Sequence-based reagent | *Il6_F* | This paper | PCR primer | CCCCAATTTCCAATGCTCTCC |

*Continued on next page*

*Continued*

| Reagent type (species) or resource | Designation | Source or reference | Identifiers | Additional information |
|---|---|---|---|---|
| Sequence-based reagent | *Il6_R* | This paper | PCR primer | GGATGGTGTTGGTCCTTAGCC |
| Sequence-based reagent | *Gapdh_F* | This paper | PCR primer | AGGTCGGTGTGAACGGATT |
| Sequence-based reagent | *Gapdh_R* | This paper | PCR primer | AATCTCCACTTTGCCACTGC |
| Sequence-based reagent | *Cyp7a1_F* | This paper | PCR primer | GCTGTGGTAGTGAGCTGTTG |
| Sequence-based reagent | *Cyp7a1_R* | This paper | PCR primer | GTTGTCCAAAGGAGGTTCACC |
| Sequence-based reagent | *Cyp8b1_F* | This paper | PCR primer | CCTCTGGACAAGGGTTTTGTG |
| Sequence-based reagent | *Cyp8b1_R* | This paper | PCR primer | GCACCGTGAAGACATCCCC |
| Sequence-based reagent | *Cyp27a1_F* | This paper | PCR primer | AGGGCAAGTACCCAATAAGAGA |
| Sequence-based reagent | *Cyp27a1_R* | This paper | PCR primer | TCGTTTAAGGCATCCGTGTAGA |
| Sequence-based reagent | *Cyp7b1_F* | This paper | PCR primer | TCCTGGCTGAACTCTTCTGC |
| Sequence-based reagent | *Cyp7b1_R* | This paper | PCR primer | CCAGACCATATTGGCCCGTA |
| Sequence-based reagent | *Cre_F* | This paper | PCR primer | GGTGAACGTGCAAAACAGGCTC |
| Sequence-based reagent | *Cre_R* | This paper | PCR primer | AAAACAGGTAGTTATTCGGATCATCAGC |
| Sequence-based reagent | *Tcrd_F* | This paper | PCR primer | CAAATGTTGCTTGTCTGGTG |
| Sequence-based reagent | *Tcrd_R* | This paper | PCR primer | GTCAGTCGAGTGCACAGTTT |
| Sequence-based reagent | *Stat3$^{flox}$_F* | This paper | PCR primer | CCTGAAGACCAAGTTCATCTGTGTTGAC |
| Sequence-based reagent | *Stat3$^{flox}$_R* | This paper | PCR primer | CACACAAGCCATCAAACTCTGGTCTCC |
| Sequence-based reagent | *Rela$^{flox}$_F* | This paper | PCR primer | GAGCGCATGCCTAGCACCAG |
| Sequence-based reagent | *Rela$^{flox}$_R* | This paper | PCR primer | GTGCACTGCATGCGTGCAG |
| Chemical compound, drug | Dextran sulphate sodium salt | Sigma-Aldrich | Cat# 42867 | |
| Chemical compound, drug | Fluorescein isothiocyanate-dextran | Sigma-Aldrich | Cat# 60842-46-8 | |
| Chemical compound, drug | Chenodeoxycholic acid | Sigma-Aldrich | Cat# C9377 | |
| Chemical compound, drug | DAPI | Sigma-Aldrich | Cat# D9542 | |
| Chemical compound, drug | PowerUp SYBR | Thermo Fisher Scientific | Cat# A25742 | |

*Continued on next page*

*Continued*

| Reagent type (species) or resource | Designation | Source or reference | Identifiers | Additional information |
| --- | --- | --- | --- | --- |
| Chemical compound, drug | Fluorosheild | Sigma-Aldrich | Cat# F6182 | |
| Commercial assay or kit | NucleoSpin RNA | Macherey-Nagel | Cat# 74106 | |
| Commercial assay or kit | Primescript 1st strand cDNA synthesis kit | Takara Bio | Cat# 6110 A | |
| Software, algorithm | Prism 9 | GraphPad | 9.0 | |

## Materials and methods

### Human studies

All studies were approved by the All India Institute of Medical Science Ethics Committee for postgraduate research (Approval number – IECPG-270/22.04.2019). Biopsy specimens were collected from recto-sigmoidal or sigmoidal colon regions. Ulcerative colitis patients with mild-to-moderate disease activity (SCCAI: 3–9) were included in the IBD group. Subjects undergoing sigmoidoscopy for inspection of manifestations such as haemorrhoidal bleeds were included in the non-IBD group. Patients with severe disease activity, a history of antibiotics or topical steroids in the past 4 weeks, pregnancy, comorbid illnesses, and/or a history of bowel surgery were excluded. These samples were immediately stored at –80°C in cryovials till further processing.

### Animal studies

All mouse strains were housed at the National Institute of Immunology (NII) and utilized adhering to the institutional guidelines (Approval number – IAEC 579/21). 5–7 week old C57BL/6 mice of a were used. Hepatocyte-specific knockout animals (Cre under albumin promoter) $Rela\Delta$hep, $Stat3\Delta$hep, and $Rela\Delta$hep$Stat3\Delta$hep along with their Cre-negative (referred to as wild-type in the text) littermates $Rela^{fl/fl}$, $Stat3^{fl/fl}$, and $Rela^{fl/fl}Stat3^{fl/fl}$ were generously gifted by Dr. Lee Quinton, School of Medicine, Boston University, Boston, MA, USA. Above mentioned knockout strains were crossed with their corresponding wild-type littermates to expand the colonies for experimental purposes at the small animal facility.

### Induction and assessment of colitis in mice

As described earlier (*Kiesler et al., 2015*), 5–7 weeks old male/female (18–21 g body weight) mice of the indicated genotypes were randomly chosen for administration with 2% of DSS in drinking water for 6 days. Subsequently, body weight and disease activity were assessed for 6 days from the onset of DSS treatment. All experiments were performed using littermate mice cohoused for a week prior to the initiation of the experiments. The disease activity index was estimated based on stool consistency and rectal bleeding. The score was assigned as follows – 0 points were given for well-formed pellets, 1 point for pasty and semi-formed stool, 2 points for liquid stool, 3 points for bloody smear along with stool, and 4 points were assigned for bloody fluid/mortality. Mice with more than 30% loss of body weight were considered moribund, and euthanized. For specific experiments, mice were euthanized at the indicated days post-onset of DSS treatment, and colon tissues were collected.

10 mg/kg body weight of chenodeoxycholic acid dissolved in DMSO was injected via intraperitoneal route into mice as described earlier (*Ward et al., 2017*).

### Histopathological studies

At day 6 of DSS treatment, mice with the indicated genotypes were euthanized, and the entire colon was excised. The colon length was measured from the rectum to the caecum. Subsequently, distal colons were washed with PBS, fixed in 10% formalin, and embedded in paraffin. 5 µm thick tissue sections were generated from the inflamed region and stained with hematoxylin and eosin (H&E). Alternately, sections were stained with Alcian Blue to reveal mucin content. Images were acquired using Image-Pro6 software on an Olympus inverted microscope under a 20 X objective lens. The

severity of colitis was assessed by epithelial damage and infiltration of inflammatory immune cells in the submucosa of the colon.

Three experimental replicates was used for histological scoring, two fields per section, and a total of three sections from each set were examined. Histological injury was assessed by a combined score of inflammatory cell infiltration (score 0–3) and mucosal damage (score 0–3) as previously described in *Ren Y* et al. 2019.

## Intestinal permeability assay

For assessing intestinal permeability, FITC-dextran was orally gavaged to untreated or DSS-treated mice 6 hr prior to sera collection. Fluorescent-based quantitation of sera samples were performed using CLARIOstar microplate reader ($\lambda_{ex}$: 490 nm and $\lambda_{em}$: 520 nm).

## Antibiotic treatment

Gut sterilization was achieved by administration of an antibiotic cocktail: ampicillin 1 g/l, neomycin 1 g/l, metronidazole 0.25 g/l, and vancomycin 0.5 g/l as described in *Hernández-Chirlaque C* et al. 2016. Antibiotic treatment was applied for 4 weeks before the start of DSS-treatment and was maintained until the end of the experiment. Depletion of gut microbiota was confirmed by conventional bacterial enumeration studies from the faaces of mice before and during the antibiotic treatment.

## Confocal microscopy: Sample preparation and analysis

### Sample preparation

Distal colon samples were fixed in 10% formalin for 24 hr. Fixed sections were embedded in paraffin and 5 μm based sections were generated using microtome. Deparaffinization of sections were achieved by snap heating followed by xylene wash, subsequently, these were rehydrated. Antigen retrieval was performed using a sodium citrate buffer (10 mM, pH 6.0) at 95° for 10 min and the slides were allowed to cool down. Furthermore, these were rinsed in PBS, permeabilized for 5 min in 0.4% PBST, and blocked for 1 hr in 5% bovine serum albumin. These sections were stained with fluorescently conjugated antibodies listed in key rescource table for overnight at 4°. Non-conjugated primary antibodies were incubated O/N at 4°s followed by incubation of conjugated secondary antibody for 2 hr. Subsequently, the slides were rinsed in PBS and incubated with DAPI for nuclear staining. Finally, slides were mounted using fluoroshield mounting media, and slides were analyzed under a ZEISS LSM 980 confocal microscope at 40 X (oil) magnification.

For analysis, the ratio between the signal of interest/DAPI was calculated for every field. Each dot represents the average ratio for five individual fields of interest. The plot is representative of data from three biological replicates.

## Gene expression studies

Total RNA was isolated using MN-NucleoSpin RNA (as per manufacturer's instructions) from the liver and colon of untreated and DSS-treated animals of indicated genotypes. cDNA was synthesized using a Takara cDNA synthesis kit as per manufacturer protocol. RT-qPCR was performed using PowerUp SYBR Green PCR master mix in the ABI 7500 FAST instrument. Relative mRNA levels were quantitated and GAPDH was used as a control.

Furthermore, RNA isolated from liver tissue was subjected to paired-end RNA sequencing after rRNA depletion, on Illumina platform 6000 at the core facility of the CCMB, India. Twenty million unique RNA fragments were sequenced per sample and the average length of the sequenced reads was around 150 bp. The RNA-seq raw data is available in the NIH National Center for Biotechnology Information GEO database as GSE243307. Quality control and sequence trimming was performed using fastp (v0.23.2). The trimmed paired-end reads were aligned to the mouse genomes (mm9) using the HISAT2 (v2.2.1) pipeline. Reads were assembled into transcripts using StringTie (v2.2.1). Annotation was conducted using aligned sequences and a GTF annotation file. The mapped reads were then used for generating the count table using StringTie (v2.2.1). We had a total of 51,610 entries in our dataset for which differential expression analysis was performed using the DEseq2 R package. Pathway enrichment was performed using the GO database and clusters were visualized using the R package ClusterProfiler.

## Biochemical studies

### Immunoblot analyses

Tissue fragments (colon or liver) were homogenized in hand held douncer in SDS-RIPA buffer (50 mM Tris-HCl pH 7.5, 150 mM NaCl, 1 mM EDTA, 0.1% SDS, 1% Triton-X 100, 1 mM DTT, 1 X Proteinase inhibitor, 1 X Protease inhibitor). Subsequently, these extracts were centrifuged and the supernatants were resolved by SDS-PAGE and transferred onto a PVDF membrane, and immunoblotting was performed using indicated antibodies. Blots were developed using Immobilon HRP substrate and images were acquired through the ImageQuant Chemiluminescent imaging system (LAS 500). Band intensities were quantified in ImageJ.

### Targeted metabolomics

Lipid pools were extracted by a two-step liquid-liquid extraction protocol as described previously (*Gaikwad, 2020*). Specific weight of the murine tissue was homogenized in methanol twice using 1.0 mm Zirconium beads. Extracted tissue pellets were subjected to chloroform extraction using a bath sonicator twice. The methanol and chloroform fractions were pooled and evaporated using speed-vac. Samples were added with 150 ul MeOH, vortexed, sonicated, and filtered (0.45 μm) for further analysis. A triple quadruple mass spectrometer (Waters, Milford, MA, USA) recorded MS and MS/MS spectra using electrospray ionization (ESI) in negative ion (NI) mode. Waters Acquity UPLC system was connected to the triple-quadrupole mass spectrometer. Analytical separations of the mixture of specific primary BAs were performed on the UPLC system using an ACQUITY UPLC C18 (1.6 mm 1×150 mm) analytical column.

### Untargeted Metabolomics

5 mg of biopsy material was transferred to pre-chilled homogenizer vials, mixed with 80% methanol, and subjected to three rounds of homogenization, each with two cycles of 30 s. The homogenate was centrifuged at 14,000 g for 20 min at 4°C and thereafter dried overnight in Speed Vac. At the time of the sample run the samples were resuspended in 15% ACN for Reverse Phase Chromatography, and LC/MS was performed on a Thermo Fisher Orbitrap Fusion Tribrid Mass Spectrometer.

## Statistical analysis

Quantified data for 3–6 mice replicates were plotted as mean ± SEM (represented as error bars). Unless otherwise mentioned, unpaired two-tailed Student's t-test and one-way Anova was used for calculating statistical significance in data sets involving two groups and multiple groups, respectively.

## Data and material availability

The mice strain used has been gifted by Dr. Lee Quinton from Boston University and is available at the breeding facility at the National Institute of Immunology. All materials and reagents will be available on request. The transcriptomic data is available online at GEO with accession id GSE243307. All codes used for the analysis are publically available and can also be provided upon request.

# Acknowledgements

We thank Dr. Lee Quinton for providing us with hepatocyte specific knockout animals for *rela* and/or *stat3* for carrying out this study; Dr. Devram S Ghorpade, NII, New Delhi for discussions; Next Generation Sequencing (NGS) facility at CSIR-CCMB for the transcriptomic support; Mass Spectrometric facility at THSTI, Faridabad for the metablomic support; the Small Animal Facility at the National Institute of Immunology, New Delhi for support with mice providing the breeding and experimental support; the Department of Biotechnology (DBT) for institutional support provided to National Institute of Immunology.

# Additional information

#### Competing interests

Soumen Basak: Reviewing editor, *eLife*. The other authors declare that no competing interests exist.

### Funding
No external funding was received for this work.

### Author contributions
 Jyotsna, Conceptualization, Data curation, Formal analysis, Validation, Investigation, Visualization, Methodology, Writing – original draft, Writing – review and editing; Binayak Sarkar, Data curation, Formal analysis, Validation, Investigation, Visualization, Methodology; Mohit Yadav, Data curation, Formal analysis, Supervision, Validation, Investigation, Visualization, Methodology; Alvina Deka, Manasvini Markandey, Priyadarshini Sanyal, Nilesh Gaikward, Investigation, Methodology; Perumal Nagarajan, Resources, Investigation, Methodology; Vineet Ahuja, Resources, Supervision; Debasisa Mohanty, Resources, Software, Supervision, Funding acquisition, Project administration; Soumen Basak, Conceptualization, Resources, Supervision, Validation, Visualization, Methodology, Writing – original draft; Rajesh S Gokhale, Conceptualization, Resources, Data curation, Supervision, Validation, Methodology, Project administration, Writing – review and editing

### Author ORCIDs
Rajesh S Gokhale  https://orcid.org/0000-0001-6597-2685

### Ethics
This study was performed in strict accordance with the recommendations in the guidelines provided by the institutional animal ethical committee. The protocol was approved by the Committee on the Animal Ethical Committee of the National Institute of Immunology (Approval number - IAEC 579/21).

Reviewer #2 (Public Review): https://doi.org/10.7554/eLife.93273.3.sa1
Reviewer #3 (Public Review): https://doi.org/10.7554/eLife.93273.3.sa2
Author response https://doi.org/10.7554/eLife.93273.3.sa3

---

## Additional files

### Supplementary files
• MDAR checklist

### Data availability
Sequencing data have been deposited in GEO under the accession code GSE243307. All data generated or analysed during the study are included in the manuscript and supporting files, source data files have been provided for Figure 1 to 5.

The following dataset was generated:

| Author(s) | Year | Dataset title | Dataset URL | Database and Identifier |
|---|---|---|---|---|
| Gokhale RS, Mohanty D, Basak S, Nandicoori VK, Singh J, Sanyal P | 2024 | A hepatocyte-specific transcriptional program driven by Rela and Stat3 exacerbates experimental colitis in mice by modulating bile synthesis | https://www.ncbi.nlm.nih.gov/geo/query/acc.cgi?acc=GSE243307 | NCBI Gene Expression Omnibus, GSE243307 |

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
