## [Editor Report · eLife assessment]

The current version of the study presents **important** findings on how the RelA/Stat3-dependent gene program in the liver influences intestinal homeostasis. The evidence supporting the conclusions is **solid**, with new data added compared to an earlier version of the study. The work will be of interest to scientists in gastrointestinal research fields.

---

## [Referee Report · Reviewer #2 (Public Review)]

Summary:

Singh and colleagues employ a methodic approach to reveal the function of the transcription factors Rela and Stat3 in the regulation of the inflammatory response in the intestine.

Strengths of the manuscript include the focus on the function of these transcription factors in hepatocytes and the discovery of their role in the systemic response to experimental colitis. While the systemic response to induce colitis is appreciated, the cellular and molecular mechanisms that drive such systemic response, especially those involving other organs beyond the intestine are an active area of research. As such, this study contributes to this conceptual advance. Additional strengths are the complementary biochemical and metabolomics approaches to describe the activation of these transcription factors in the liver and their requirement - specifically in hepatocytes - for the production of bile acids in response to colitis.

In this revised version, the authors have addressed previously raised questions.

---

## [Referee Report · Reviewer #3 (Public Review)]

Summary:

The authors try to elucidate the molecular mechanisms underlying the intra-organ crosstalks that perpetuate intestinal permeability and inflammation.

Strengths:

This study identifies a hepatocyte-specific rela/stat3 network as a potential therapeutic target for intestinal diseases via the gut liver axis using both murine models and human samples.

Weaknesses:

(1) The mechanism by which DSS administration induces the activation of the Rela and Stat3 pathways and subsequent modification of the bile acid pathway remains clear. As the authors state, intestinal bacteria are one candidate, and this needs to be clarified. I recommend the authors investigate whether gut sterilization by administration of antibiotics or germ free condition affects 1. the activation of the Rela and Stat3 pathway in the liver by DSS-treated WT mice and 2. the reduction of colitis in DSS-treated relaΔhepstat3Δhep mice.

(2) It has not been shown whether DSS administration causes an increase in primary bile acids, represented by CDCA, in the colon of WT mice following activation of the Rela and Stat3 pathways, as demonstrated in Figure 6.

(3) The implications of these results for IBD treatment, especially in what ways they may lead to therapeutic intervention, need to be discussed.

The above weakness points have been resolved by the revision and additional experiments.

---

## [Author Response]

The following is the authors’ response to the original reviews.

**eLife assessment**
This important study reveals the RelA/Stat3-dependent gene program in the liver influences intestinal homeostasis. The evidence supporting the conclusions is compelling, although some additional experiments will strengthen the study. The work will be of interest to scientists in gastrointestinal research fields.
**Public Reviews:**

**Reviewer #1 (Public Review):**
Summary:In this study, the authors showed that activation of RelA and Stat3 in hepatocytes of DSS-treated mice induced CYPs and thereby produced primary bile acids, particularly CDCA, which exacerbated intestinal inflammation.Strengths:This study reveals the RelA/Stat3-dependent gene program in the liver influences intestinal homeostasis.

Our reply: We thank the reviewer for the positive feedback and for appreciating the strength of our study.

Weaknesses:Additional evidence will strengthen the conclusion.(1) In Fig. 1C, photos show that phosphorylation of RelA and Stat3 was induced in only a few hepatocytes. The authors conclude that activation of both RelA and Stat3 induces inflammatory pathways. Therefore, the authors should show that phosphorylation of RelA and Stat3 is induced in the same hepatocytes during DSS treatment.

Our reply: The reviewers have raised a pertinent issue in Figure 1, as later on in our study we suggest that the combined activation of Rela and Stat3 is critical for aggravating the colitogenic phenotype in the murine model.

To address this issue, we have co-stained the fixed liver tissue of untreated and DSS-treated wild type mice with p-RelA (Ser536) and p-Stat3(Ser727) antibodies. Author response image 1 below shows the single staining for p-Rela (Ser536), pStat3 (Ser727), DAPI (to demarcate the nuclei) and merged image (p-Rela + pStat3).

Further, the signal intensity of p-RelA (Ser536) and p-Stat3(Ser727) per nuclei was calculated and plotted as a box plot. It is evident that the median of p-Rela and p-Stat3 signal intensity in DSS-treated samples is more than that of the control samples, suggesting that the majority of the treated hepatocytes have the presence of both p-Rela and p-Stat3 in the nuclei.

**Author response image 2. sa3fig2:** 

Further, we calculate the number of nuclei in the DSS-treated samples which are above the 90th percentile of the control samples (data has been provided in Author response table 1 below). We also calculate the percentage overlap of p-Rela to p-Stat3 and vice versa in Author response table 1 below.

**Author response table 1. sa3table1:** 

	The totalnumber ofnucleianalyzed	90th percentile ofsignal intensity forcontrol samples	Nuclei of DSS-treatedsections with signalintensity above the 90thpercentile	Percentageoverlap
For Relaprobedsamples	29	3.465	27	100
For Stat3probedsamples	29	2.935	29	93

Together our analysis concludes that indeed there is an activation of Rela and Stat3 in the same hepatocytes to generate the downstream effect that we observe in our study post-DSS treatment.

(2) In Fig. 5, the authors treated mice with CDCA intraperitoneally. In this experiment, the concentration of CDCA in the colon of CDCA-treated mice should be shown.

Our reply: We have experimentally examined if the CDCA supplemented intraperitoneally at the experimental dose used in our study, is reaching the colon or not. To quantify colonic CDCA we have performed targeted mass spectrometric studies and the data has been provided as a bar plot below.

**Author response image 3. sa3fig3:** 

It is evident from the plot that the CDCA levels are significantly higher in mice supplemented with CDCA as compared to their corresponding control (where only the vehicle was supplemented). The data has been added to the supplementary section S5b and the main text has been modified accordingly.

**Reviewer #2 (Public Review):**
Singh and colleagues employ a methodical approach to reveal the function of the transcription factors Rela and Stat3 in the regulation of the inflammatory response in the intestine.Strengths of the manuscript include the focus on the function of these transcription factors in hepatocytes and the discovery of their role in the systemic response to experimental colitis. While the systemic response to induce colitis is appreciated, the cellular and molecular mechanisms that drive such systemic response, especially those involving other organs beyond the intestine are an active area of research. As such, this study contributes to this conceptual advance. Additional strengths are the complementary biochemical and metabolomics approaches to describe the activation of these transcription factors in the liver and their requirement - specifically in hepatocytes - for the production of bile acids in response to colitis.

Our reply: We express our gratitude to the reviewer for recognizing and appreciating the mechanistic insight provided by our work, and for considering it valuable in advancing conceptual understanding in the relevant field.

Some weaknesses are noted in the presentation of the data, including a comprehensive representation of findings in all conditions and genotypes tested.

Our reply: We thank the reviewer for the query and we have suitably modified the figures for a comprehensive representation of the findings, as described below:

● In Figure 2C, we have added the control alcian blue stained samples to clarify that there were no qualitative differences in the mucin levels observed in the *relaΔhepstat3Δhep* as compared to the wild type mice.

● We have also modified the figure 2D for a better presentation of the data.

● We have included histopathological analysis for the *relaΔhepstat3Δhep* mice in Figures S3a and S3b, following a format similar to the wild-type data previously provided as Figure S1a and S1b.

● For Figure 5C, the corresponding untreated samples with and without CDCA supplementation have been provided in the supplementary section Figure S5e.

● For Figure 2E, 3E, and 4C - the RT-qPCR data of the DSS-treated samples is plotted relative to their corresponding control samples, hence we only display two conditions in the bar plot. We have accordingly modified the figure legend for better clarity.

**Reviewer #3 (Public Review):**
Summary:The authors try to elucidate the molecular mechanisms underlying the intra-organ crosstalks that perpetuate intestinal permeability and inflammation.Strengths:This study identifies a hepatocyte-specific rela/stat3 network as a potential therapeutic target for intestinal diseases via the gut-liver axis using both murine models and human samples.

Our reply: We thank the reviewer for appreciating the therapeutic potential of our work.

Weaknesses:(1) The mechanism by which DSS administration induces the activation of the Rela and Stat3 pathways and subsequent modification of the bile acid pathway remains clear. As the authors state, intestinal bacteria are one candidate, and this needs to be clarified. I recommend the authors investigate whether gut sterilization by administration of antibiotics or germ-free condition affects 1. the activation of the Rela and Stat3 pathway in the liver by DSS-treated WT mice and 2. the reduction of colitis in DSS-treated relaΔhepstat3Δhep mice.

Our reply: We thank the reviewer for bringing up the aspect of gut microbiota in imparting colitis in our mice model. In accordance with reviewer's recommendation, we have sterilized the gut by administration of antibiotics, to evaluate if the intestinal bacteria are an important component leading to the activation of Rela and Stat3 pathway in the liver of DSS-treated WT mice or not.

(a) A brief schematic representation of the experimental design has been provided below and the detailed description of the methods has been described in supplementary methods.

**Author response image 4. sa3fig4:** 

Extract of liver tissues from mice treated with DSS for 6 days with/without prior antibiotic treatment were probed with p-Stat3 (Ser727) to examine the activation status of the hepatic Stat3 pathway. We observe that the signals for p-Stat3 (Ser727) are comparatively reduced post antibiotic treatment as evident from the blot below. p-Stat3 (Ser727) was a prominent activation signal at Day 6 DSS treatment that we have observed in Figure 1D,E.

**Author response image 5. sa3fig5:** 

These studies suggest that the activation status of Stat3 activation is hampered by antibiotic treatment and considering that Rela and Stat3 have to coordinate activity, presumably the downstream activation will be modulated upon gut sterilization. However, it should be appreciated that a sterilized gut is not likely to be physiologically relevant and intestinal bacteria along with bile acid levels would modulate Rela/Stat3 pathways.

b) It is likely that the hepatic deficiency of Rela and Stat3 may have modified the gut microbiome in *relaΔhepstat3Δhep* mice because of the altered bile composition. Moreover, the gut microbiota is a key component that guides the outcome of colitis. Hence, future studies are important to examine the role of the gut microbiome in imparting resistance in *relaΔhepstat3Δhep* mice, to colitogenic insults.

(2) It has not been shown whether DSS administration causes an increase in primary bile acids, represented by CDCA, in the colon of WT mice following activation of the Rela and Stat3 pathways, as demonstrated in Figure 6.

Our reply: In order to address the query, we would kindly like to request the reviewers to look at figure 4B where we show an increase in the CDCA levels of the colonic tissue, which is corresponding to our CDCA levels in the liver tissue (figure 4A) thus indicating that it may be driven by the hepatic Rela and Stat3 pathways.

(3) The implications of these results for IBD treatment, especially in what ways they may lead to therapeutic intervention, need to be discussed.

Our reply: We are grateful to the reviewer for bringing this topic for discussion.

Until now, only immunosuppressive agents and immunomodulators have been conventionally considered as therapeutic measures to manage IBD. However, with increasing research on the role of hepatic bile acid metabolism during experimental colitis, its potential cannot be undermined in the clinical setting. The potential of bile acids as a therapeutic target has been harnessed in the past; bile acid sequestrants have been utilized as a treatment for hyperlipidemia 46. Remedies like fecal microbial transplantation, which serve to normalize the bile acid ratios in the gut, are emerging as potential therapeutics in the last decade for IBD 47, 40. However, the potential of altering hepatic bile metabolism has remained unexplored for IBD, possibly due to a lack of mechanistic insight. Towards this, our work demonstrates the pro-inflammatory potential of CDCA during colitis following the activation of the Rela/Stat3 pathway. The suppression of Rela/Stat3-induced CDCA could provide beneficial effects in IBD patients while protecting the basal bile acid levels (through FXR signaling). Thus our studies identify a hepatocyte-specific rela/stat3 network as a potential therapeutic target for intestinal diseases. Another approach could be the use of bile acid sequestrants, which will temporarily decrease the levels of primary bile acids in the colon until the proinflammatory pathways are dampened as a combinatorial therapy alongside existing treatments.

**Recommendations for the authors:**

**Reviewer #1 (Recommendations For The Authors):**
Minor:Fig. 4C should be Fig. 4D and vice versa.

Our reply: We have swapped Fig. 4C and Fig. 4D and corresponding changes have been incorporated in the main text.

**Reviewer #2 (Recommendations For The Authors):**
Please make note of the following specific commentsThe immunostainings for phosphorylated p-Rela and STAT3 are unclear. Is there nuclear translocation of these phosphorylated transcription factors? Can the authors enumerate the percentage of cells in which nuclear translocation (presumably in hepatocytes) is detected?

Our reply: We apologize that immunostainings for phosphorylated p-Rela and STAT3 are unclear to the reviewers. Here we have tried our best to make the data clear by analyzing the stained section and plotting them.

To start with, we have co-stained the fixed liver tissue of untreated and DSS-treated wild type mice with p-RelA (Ser536) and p-Stat3(Ser727) antibodies, below we have provided a representative image used for analysis. To demarcate the nuclear boundary of the hepatocytes DAPI was used and the signal intensity for p-RelA (Ser536) and p-Stat3(Ser727) was quantified using ZenBlue software.

**Author response image 6. sa3fig6:** 

Below we have provided the box plot for the calculated nuclear intensities in the control (untreated) and DSS-treated samples for p-Rela and p-Stat3. We can clearly see that the median of p-Rela and p-Stat3 signal intensity in DSS-treated samples is more than that of the control samples, suggesting that the majority of the treated hepatocytes have the translocation of p-Rela and p-Stat3 in their nuclei.

**Author response image 7. sa3fig7:** 

The figure legends for Figures 2C and D are flipped. Please correct.

Our reply: Thank you for pointing it out, our apologies for the error and we have corrected the figure 2 accordingly.

For all H&E stainings, the authors should include histological scoring disease severity.

Our reply: Thank you for the query put forward, histological scoring to quantify the qualitative data obtained through microscopy is given below. Dot plot for the histological scoring of the H&E data for untreated and DSS-treated colon samples, we have referred to the scale described by *Ren Y et al.* 2019 (doi: 10.1038/s41598-019-53305-z) to score the sections.

**Author response image 8. sa3fig8:** 

We have added the dot plot to supplementary figure 2d, also the method applied for the above analysis has been described in the supplementary method section.

Please include Alcian Blue Staining in non-DSS treated WT and rel/stat3 double cKO mice.

Our reply: Thank you for pointing this out, we have added the Alcian Blue Staining of non-DSS treated WT and rel/stat3 double KO mice to figure 2C

For Figure 3C, can the authors indicate in the figure itself which bile acid is being represented (not only in the Figure legend)?

Our reply: Thank you for the suggestion we have indicated the respective bile acid in Figure 3C for better understanding.

As these data are from untargeted metabolomics, were other bile acids detected?

Our reply: This is a part of a separate study conducted by our collaborator, and will form a part of a new manuscript which will be focussed on human studies.

Can the authors validate the downregulation of key enzymes shown in Figure 3D, E at the protein level?

Our reply: We agree with the reviewer’s comment, that mRNA levels are not critical determinants of activation of any pathway, rather an indicator of probable activation. In that scenario, the estimation of protein levels is more determinative. But taking into consideration that we have the metabolomic data in subsequent figures (as in Figure 4 A, B) supporting our findings in Figure 3D, E, this makes RT-qPCR data a more robust indicator of an activated hepatic bile acid biosynthesis machinery.

The figure legends for Figures 4C and D are flipped. Please correct.

Our reply: Taking into consideration the suggestions by reviewer 1 we have swapped Fig. 4C and Fig. 4D and corrected the legend placement accordingly, thank you for pointing this out.

Also, please include representative images for the data represented in 4C.

Our reply: Thank you for the query, we have already added the representative images of confocal microscopy as figure S4.

Figure 5B should indicate that the data presented is from double cKO mice.

Our reply: We have indicated that the colon length data is from double KO animals in figure to make the visual representation clear for the readers, thank you for the concern.

Please correct typos: "entrocytic" and "Untread" in Figure Legend 5.

Our reply: Thank you for pointing out the error in the Legend, we apologize for the error in these errors we have corrected Figure 5.

Figure S4 includes a dataset (qPCR for Mmp3) that is not described. Neither Figure S4 nor S5 are described in the text.

Our reply: Thank you for the query, firstly we have already added Figure S4 and S5 to the text, our apologies that it has not been properly highlighted.

Secondly, the data for RT-qPCR for Mmp3 has been removed from supplementary figures as it may not be very relevant to the study.

Overall, the manuscript should be edited to ensure the correct use of English. Please also note that the last name of the first author seems to be missing in the main text.

Our reply: Thank you for the suggestion we have re-checked the manuscript for the probable errors and rectified them. The first author has a single name (with no surname) and we would like to correct that during the final print of the manuscript.

**Reviewer #3 (Recommendations For The Authors):**
(1) The authors need to show if DSS treatment affects the serological or histological changes in the liver of relaΔhepstat3Δhep mice.

Our reply: To address that, we have analyzed key serological markers of liver damage as well as looked into tissue histology.

The pathophysiological parameters of the liver of DSS treated *relaΔhepstat3Δhep* mice has been added to the revised manuscript as figure S3a and S3b. Here we show that the serological parameters are within the physiological range upon DSS treatment (Author response image 9a). Besides, the histological parameters remain unaltered as compared to the control tissue (Author response image 9b).

Cumulatively, both at the tissue level and functional level, there is not much effect of DSS

treatment on liver of *relaΔhepstat3Δhep* mice.

**Author response image 9. sa3fig9:** 

(2) It is recommended to use a second model to verify if this phenomenon is applicable to colitic status in general.

Our reply: We appreciate the query put forward, this is an ongoing study and we hope to examine further the role of hepatic RelA and Stat3 in TNBS-induced colitis model and in T cell transfer model of colitis.